# Printing the Ultra-Long Ag Nanowires Inks onto the Flexible Textile Substrate for Stretchable Electronics

**DOI:** 10.3390/nano9050686

**Published:** 2019-05-02

**Authors:** Sheng-Hai Ke, Qing-Wen Xue, Chuan-Yuan Pang, Pan-Wang Guo, Wei-Jing Yao, He-Ping Zhu, Wei Wu

**Affiliations:** 1Research Center of Intelligent Packaging, School of Packaging Design and Art, Hunan University of Technology, Zhuzhou 412007, China; shenghaiK@outlook.com (S.-H.K.); chuanyuanpang@163.com (C.-Y.P.); gpw21@outlook.com (P.-W.G.); hepingzhu@sina.com (H.-P.Z.); 2National & Local Joint Engineering Research Center of Advanced Packaging Materials Developing Technology, Hunan University of Technology, Zhuzhou 412007, China; 3Laboratory of Printable Functional Nanomaterials and Printed Electronics, School of Printing and Packaging, Wuhan University, Wuhan 430072, China; qingwenxue@whu.edu.cn (Q.-W.X.); weijingyao25@163.com (W.-J.Y.); 4Shenzhen Research Institute of Wuhan University, Shenzhen 518057, China

**Keywords:** stretchable electronics, Ag nanowires, screen printing, flexible electronics, printed electronics

## Abstract

Printing technology offers a simple and cost-effective opportunity to develop all-printed stretchable circuits and electronic devices, possibly providing ubiquitous, low-cost, and flexible devices. To successfully prepare high-aspect-ratio Ag nanowires (NWs), we used water and anhydrous ethanol as the solvent and polyvinylpyrrolidone (PVP) as the viscosity regulator to obtain a water-soluble Ag NWs conductive ink with good printability. Flexible and stretchable fabric electrodes were directly fabricated through screen printing. After curing at room temperature, the sheet resistance of the Ag NW fabric electrode was 1.5 Ω/sq. Under a tensile strain of 0–80% and with 20% strains applied for 200 cycles, good conductivity was maintained, which was attributed to the inherent flexibility of the Ag NWs and the intrinsic structure of the interlocked texture.

## 1. Introduction

As a newly developed technology, flexible and stretchable electronics are achieving a great variety of applications because they can be compressed, twisted, and conformed to a complex non-planar surface [1,2]. With the increasing use of wearable intelligent electronic devices, fabric flexible electrodes have attracted more and more attention [3,4,5,6]. Owing to the particularity of the textile substrate, the conventional silicon-based microelectronic device manufacturing technology is not suitable for the low-cost mass preparation of textile-based flexible electronics [7]. In contrast, printed electronic technology uses different printing methods to pattern materials with good dispersibility or solubility in liquid, to achieve the preparation of electronic components. Printed electronic technology has many advantages, including low energy consumption, low consumables, green environmental protection, flexibility, and low equipment investment. The easy manufacturing process allows the large-scale mass production of traditional electronic products [8,9,10,11]. Studies in printed electronics involve many subjects including common technologies in materials, equipment, processes, and applications [12]. In the field of printed electronics, one study focused on the preparation of metallic nanomaterial-based conductive inks, because metal nanomaterials have small size, low sintering temperature, and an easy process of ink formulation.

Currently, metallic nanoparticles (NPs)-based conductive inks are most commonly used in printed electronics, where the conductive components are mostly silver, copper, and copper–silver nanoparticles [13,14,15,16]. The conductive inks require thermal sintering treatment to fuse the nanoparticles to form a conductive path after printing. The sintering temperature of metallic NPs-based conductive inks is generally above 150 °C. However, the high temperature resistance of the conventional textile is poor, therefore Ag NPs-based inks are not suitable for the preparation of soft textile electrodes. Generally, Ag nanowires (NWs) possess a high aspect ratio, and only a small number of resistive contact points are formed when constructing a conductive network, therefore conductive paths can be formed without high-temperature sintering. It can be seen that conductive inks based on Ag NWs have the ability to be printed and serve as textile electrodes. However, few studies have successfully formulated Ag NWs conductive inks with good printability and high conductivity, mainly because Ag NWs agglomerate more easily than Ag NPs [17]. Some stabilizers or resin additives are suitable for Ag NPs conductive ink but do not allow a good dispersion of Ag NWs in the solvent. The preparation of long Ag NWs with uniform morphology using pure products and the formulation of Ag NWs ink using a low sintering temperature remain a challenge.

Recently, it was reported that deionized water was used as a solvent and hydroxypropyl methylcellulose (HPMC) was used to adjust the viscosity of the ink to formulate Ag NW inks, which were printed onto the surface of paper and polyethylene terephthalate (PET). The nanowire conductive ink had good electrical conductivity (the conductivity is 4.67 × 10^4^ S/cm) after thermal sintering at 150 °C [18]. However, this sintering temperature is still relatively high for the textile substrates. Therefore, it has been a challenge to develop an Ag NWs conductive ink with the synergistic properties of good printability, low sintering temperature, and high electrical conductivity. In this study, a water-soluble Ag NWs conductive ink with good printability was formulated by preparing pure Ag NWs with a high aspect ratio using water and absolute ethanol as the solvent, and polyvinylpyrrolidone (PVP) as the binder and viscosity controller. This conductive ink was screen-printed directly onto a soft stretchable textile, which maintained good electrical conductivity under the tensile strain of 0–80% and with stretching of 20% for 200 cycles. The conductive mechanism of the printed tracks was investigated.

## 2. Methods

### 2.1. Materials and Chemicals

Silver nitrate (AgNO_3_, AR) and absolute ethanol (C_2_H_5_OH, AR) were purchased from Sinopharm Chemical Reagent Co., Ltd (Beijing, China). Copper (II) chloride dehydrate (CuCl_2_·2H_2_O, AR) was purchased from Shanghai Aladdin Reagents Co., Ltd (Shanghai, China). PVP (MW ≈ 360000) was purchased from Sigma-Aldrich (Shanghai, China). Deionized water (18.2 MΩ) was used in all experiments.

### 2.2. Formulation of Ag NWs Inks

According to our previous report [19], Ag NWs are fabricated by a wet-chemical method. To formulate the screen printing conductive inks, the Ag NWs were first dispersed in absolute ethanol to a mass fraction of 5.27% to form a solution A; PVP with the molar concentration of 1 mol/L was dissolved in absolute ethanol and deionized water at a volume ratio of 1:1 to form solution B. Then, according to a specific mass ratio, solution B was added to solution A, vigorously stirred, and ultrasonically shaken for 5–7 min until it was uniformly mixed to form an Ag NWs conductive ink. Subsequently, the prepared Ag NWs conductive ink was printed onto the surface of the textile and conventional paper using a 300-mesh screen printing plate and allowed to dry naturally.

### 2.3. Characterization

The rheological properties of the ink were tested using a Malvern Kinexus pro rotary rheometer with a parallel plate radius of 20 mm and an interpolate gap of 1 mm. The performance tests were carried out at room temperature (25 °C). In the process of screen printing, the shear rate of the ink was not constant. When the screen printing process was simulated, the shear rate applied to the ink was 0.1 s^−1^, which was maintained for 20 s before the ink was transferred. During the squeegeeing process, the shear rate applied to the ink was 200 s^−1^ for 30 s, and after the squeegee was completed, the shear rate applied to the ink was restored to 0.1 s^−1^ for 200 s. The oscillating rheological test was carried out under an oscillating stress of 0.1 to 100 Pa, and the frequency was 1 Hz. The surface morphology of the printed patterns was characterized by a field emission scanning electron microscope (FESEM, Hitachi S-4800) at a working voltage of 5 kV. The electrical properties of the flexible electrode were measured by a Keithley 2400 Series source meter instrument, and the electrodes resistance at different tensile strains and cyclic strains was also measured. The square resistance was measured by a four-probe DC low-resistance tester (Zhuhai Kaiwei Optoelectronics Technology Co., Ltd., FP-001).

## 3. Results and Discussion

A conductive ink is usually composed of a conductive component, a solvent, and a plurality of additives. The conductivity of the printed tracks is related to the content, dispersibility, and specific surface area of the conductive component in the ink. The solvent and additives are used to promote the wettability of the conductive particles, the stability of the dispersion, and the film-forming properties after printing [20]. In addition, the viscosity of the conductive ink needs to be tailored for screen printing. In this study, in the process of formulating the ink, the conductive component was Ag NWs with a high aspect ratio of about 500. The solvent was deionized water and absolute ethanol, which offer green environmental protection and the advantage of a low cost. The additive was PVP with an average molecular weight of 360,000. PVP is a water-soluble polymer with good film formation, cohesiveness, and solubilization. The higher the molecular weight, the higher the viscosity, thus it was used as a rheological additive to adjust the thixotropic properties of the screen printing ink. Herein, the Ag NWs synthesized by the polyol method possessed a layer of coated PVP on the surface, which effectively prevented the agglomeration and precipitation of the Ag NWs, ensuring the stable dispersion of the Ag NWs in the solvent. Thereby, the Ag NWs conductive ink that was prepared is non-toxic, and the current ink system is simple and environmentally friendly. The stability of the prepared ink is important in practical application. We rechecked the as-prepared Ag NW ink after one month; no phenomenon was observed in any layer because the ink was stored in the refrigerator.

Figure 1 shows the SEM images of the as-prepared Ag NWs; the length of the Ag NWs was around 70 μm, the average diameter was about 132 nm, and the aspect ratio was more than 500. Our previous study demonstrated that the high-molecular-weight PVP can be strongly chemisorbed on the (100) crystal plane to passivate it, which promotes one-dimensional axial growth of Ag NWs [19].

The rheological properties of the ink were measured by a flat rheometer. As shown in Figure 2a, the changes in viscosity of the ink at three different stages were measured. The shear rate of the first stage was 0.1 s^−1^ for 30 s, the shear rate of the second stage increased to 200 s^−1^ for 30 s to simulate the squeegee process during screen printing, and the third stage shear rate was reduced to 0.1 s^−1^ for 200 s to simulate the recovery of viscosity after the inks were transferred to the print substrate. The viscosity of the ink stabilized at 33.0 Pa·s at 26 s (shear rate of 0.1 s^−1^). When the shear rate increased to 200 s^−1^, the viscosity of the ink dropped to 0.3 Pa·s, ensuring the ink smoothly passed through the screen mesh when it was squeegeed and transferred to the surface of the substrate. The low viscosity also improved the resolution of the printed tracks. When the ink squeegee process finished, the shear rate was restored to 0.1 s^−1^ and the viscosity of the ink gradually recovered to 61.5 Pa·s at 70 s and to 82.87 Pa·s at 110 s. In this process, the ink was evenly distributed and formed the printed film. The viscosity of the ink after recovery was greater than the initial viscosity, probably due to the evaporation of anhydrous ethanol in the solvent.

Moreover, the oscillating rheological test was used to study the viscoelasticity of the Ag NWs ink. Different printing methods require different shear rates. In screen printing, the shear rate range of 0.1 s^−1^ to 200 s^−1^ simulates the ink transfer process during screen printing [20,21,22]. Figure 2b shows the change of the storage modulus (G’) and the loss modulus (G”) of the ink with shear stress, which can be divided into three regions [23]. The first region is the linearity of the viscoelasticity of the ink, which is the largest deformation that can be applied to the ink without changing its structure. In this region, the Ag NWs contacted each other to form a tight network structure which was resilient under pressures and tensions. Therefore, the structure of the ink in this region was not damaged by external shear stress; the shear stress was about 1.1 Pa, and the ratio of the storage modulus to loss modulus (G”/G’) of the ink was about 0.38. In the second region, the G’ and G” continuously decreased when the shear stress increased and the structure of the ink was gradually destroyed, but the storage modulus was still larger than the loss modulus, and the ink exhibited solid behavior. The third region started at the intersection of the storage modulus and the loss modulus (the shear stress was about 7.8 Pa). When the shear stress was further increased, the loss modulus of the ink gradually became greater than the storage modulus, and the inks’ behavior gradually changed to fluid and could not rely on the storage modulus to return to the initial state. If the corresponding shear stress at the junction is too large, the ink will stick and block the mesh [24]. Figure 2c shows the ratio of G”/G’ as a function of shear stress. A smaller ratio of G”/G’ means that the viscosity of the ink is large and the ink will not easily pass through the screen mesh. In contrast, if the G”/G’ ratio is too large, the viscosity of the ink is very small, and the ink will easily spread after passing through the printing plate, resulting in a decrease in the printing resolution of the tracks [25]. Figure 2d is the photograph of the as-prepared Ag NWs ink, which demonstrates that the ink possessed a good viscoelasticity. In this ink system, the good rheological properties of the ink were closely related to the high aspect ratio of the Ag NWs and the large average molecular weight of PVP.

As shown in Figure 3a,b, the formulated Ag NWs conductive ink was screen-printed onto the surface of the fabric and paper to form conductive tracks. The sheet resistance of the electrode printed on the surface of the fabric and paper was 1.5 Ω/sq and 0.7 Ω/sq, respectively. As shown in Figure 3c,d, Ag NWs were deposited on the fabric surface and in the middle of the fabric to form dense Ag NWs conductive networks. Because the fiber diameter of the paper was small, the surface of the formed Ag NWs film was flatter. There was no obvious Ag NWs bundle or agglomeration on the surface, indicating that the Ag NWs were uniformly deposited on the surface of the paper to form a permeable conductive network. Because of the high aspect ratio of the Ag NWs, the adjacent Ag NWs overlapped in the conductive network, the junction resistance was small, and the conductive path was directly formed. Therefore, the printed conductive tracks had good conductivity when dried at room temperature.

Table 1 compares the electrical conductivity of the as-prepared screen printing ink with the Ag nanoparticles (NPs), Ag flakes, and Ag NWs used as conductive components in previous studies. Although the listed conductive inks have better electrical conductivity than our Ag NWs inks, these inks require a high temperature sintering treatment (e.g., higher than 300 °C) to allow the NPs, flakes, and NWs to fuse to form a conductive path [20,26,27,28,29,30,31]. However, this sintering temperature limits the diversity of the printed substrates and is not suitable for flexible substrates such as paper, textile, and plastics. In addition, the content of the conductive component in the as-prepared Ag NWs ink is only 3.7%, which is far lower than the content of the mentioned Ag NPs or flakes inks, which means that the cost of the Ag NWs conductive ink is greatly reduced. The low content of the conductive component in the Ag NWs conductive ink is attributed to the high aspect ratio of the Ag NWs and the volatilization of the solvent during printing. It can be seen that the as-prepared Ag NWs conductive ink has the advantages of low conductive component content and good electrical conductivity after curing at room temperature, which makes it suitable for the preparation of flexible electrodes by printing inks onto temperature-sensitive substrates.

As shown in Figure 4, the interface between the printed and un-printed area, the defect area at the beginning of printed tracks, and the top-view and cross-view morphologies of the printed textile were further characterized by SEM. After screen printing, the ink penetrated from the surface of the textile into the gap between the fabric fibers (Figure 4b). This infiltration process is called the “wicking effect” and occurs in the middle of the fiber bundle. Researchers have studied the fabric and fluid transfer behavior in textiles [36,37,38]. When the ink is transferred to the surface of a textile, it penetrates into the wool of the fabric under the capillary action of the fiber bundles in the yarn. When the solvent in the ink is dried, the ink is fixed onto the surface of the fabric and no longer penetrates. As shown in Figure 4a–c, the Ag NWs formed a stretchable conductive path in the fabric yarn in accordance with the inherent braid structure of the yarn. As shown in Figure 4d, the fibers were filled with conductive Ag NWs composite materials, and the micro-/nanoscale Ag NWs penetrated into the tiny voids in the fiber bundle. However, the large gap between the fiber bundles was not filled by the conductive ink, because the Ag NWs conductive ink continued to flow and penetrate and eventually penetrated into the individual fiber bundles, which ensured the inherent structure of textile. The periodic interlocking ring-structure was not destroyed and maintained the soft and stretchable properties of the textile [30].

Because the textile substrate is intrinsically a stretchable material, tensile stress tests and cyclic strain tests were carried out on the textile electrode. The printed tracks used in the test were 15 mm long and 5 mm wide. In the tensile strain test, the tensile strain was sequentially increased from 10%, in increments of 10%, to 80%, and the stretching interval was 10 s each time. As shown in Figure 5a, the printed tracks on the textile surface were stretched; when the tensile strain reached 80%, the resistance increased from 2.8 Ω to 15.4 Ω, which was 5.5 times greater than the initial resistance. In the cyclic strain test, the resistance changes that resulted from cyclic stretching, for 200 repeats at 10%, 20%, and 30% tensile strain were tested, as shown in Figure 5b. Under the cyclic strain test, the average rate of change of line resistance was about 2.4, 3.0, and 9.3 Ω at 10%, 20%, and 30% tensile strain, respectively. The above results demonstrated that the printed tracks on the surface of the textile had good electrical conductivity under both tensile testing and cyclic strain testing. This was mainly due to the inherent stretchability of the Ag NWs and the flexibility of the weave structure (Figure 4). Since the Ag NWs conductive ink combined well with the textile, when subjected to tensile strain, the stress could be released by deformation of the woven fabric, without causing breakage of the conductive track. In addition, the Ag NWs conductive ink selectively filled small gaps in the woven fiber bundle but not large gaps, which means the textile could naturally straighten, ensuring that fabric fibers and printed tracks could simultaneously adapt to externally applied stresses. Figure 5c,d are the SEM images of the printed tracks on the surface of the textile before and after stretching at 80% strains. The conductive path formed by the woven structure of the textile showed no signs of breakage, but the gap between the bundles of the woven fabric became larger. The portion of the Ag NWs that originally filled the gap decreased, and the number of the conductive paths was reduced, thereby the resistance of the printed tracks increased. It is very difficult to prevent changes of the resistance of the printed electrode when a textile is stretched, because some conductive networks are broken, and the distance between the conductive fillers is increased [39]. The resistance cannot be fully recovered when the textile is released to a 0% strain, which may be due to the reorientation and increase in initial distance of the conductive fillers [40,41]. Our previous work reported a method where conductive fillers were embedded into a flexible substrate, which prevented the resistance change to some extent [42].

## 4. Conclusions

In summary, using PVP as a viscosity modifier and water and absolute ethanol as the solvent, an Ag NWs conductive ink with a low silver content was prepared, and the thixotropic properties and oscillating rheological properties of this ink were investigated. Subsequently, the Ag NWs conductive ink was printed onto the surface of a stretchable textile and paper by a facile screen printing technology. Under ambient drying conditions, the sheet resistance of the printed electrodes of the fabric and paper was 1.5 Ω/sq and 0.7 Ω/sq, respectively. In addition, the printing conductive electrode of the textile was further investigated, showing that the electrode maintained good electrical conductivity under the tensile strains of 0–80% and with stretching of 20% strains for 200 repeats of the cyclic test. The good stretchability of the printed electrode came from the synergistic effect of the inherent stretchability of Ag NWs and the unique deformability of fabric structures. We envision this study lays a foundation for the research and preparation of smart wearable electronic devices.

## Figures and Tables

**Figure 1 nanomaterials-09-00686-f001:**
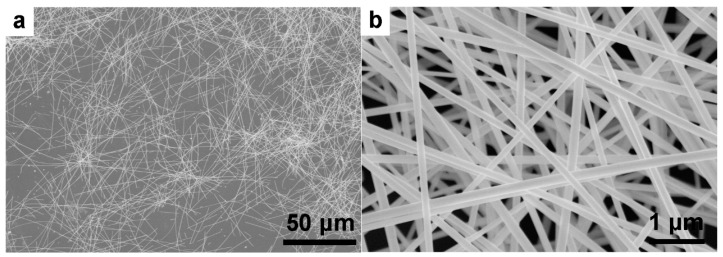
SEM images of the as-prepared Ag nanowires at different magnification: (**a**) low-magnification, (**b**) high-magnification.

**Figure 2 nanomaterials-09-00686-f002:**
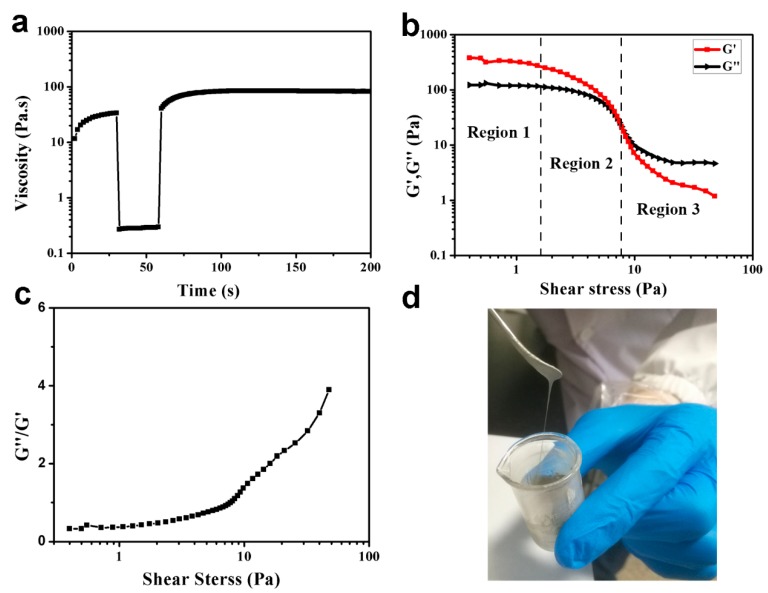
The rheological properties (**a**–**c**) and photograph (**d**) of the as-prepared Ag NWs ink.

**Figure 3 nanomaterials-09-00686-f003:**
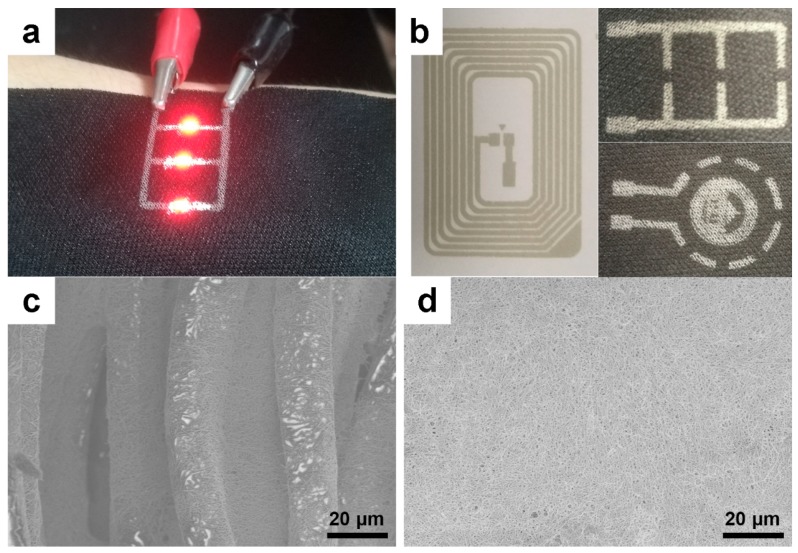
The photograph (**a**,**b**) of the screen printing patterns onto the textile (the LED can light on under the external applied voltage) and conventional paper, SEM images at different magnifications of the surface of printed layer (**c** is the textile and **d** is paper).

**Figure 4 nanomaterials-09-00686-f004:**
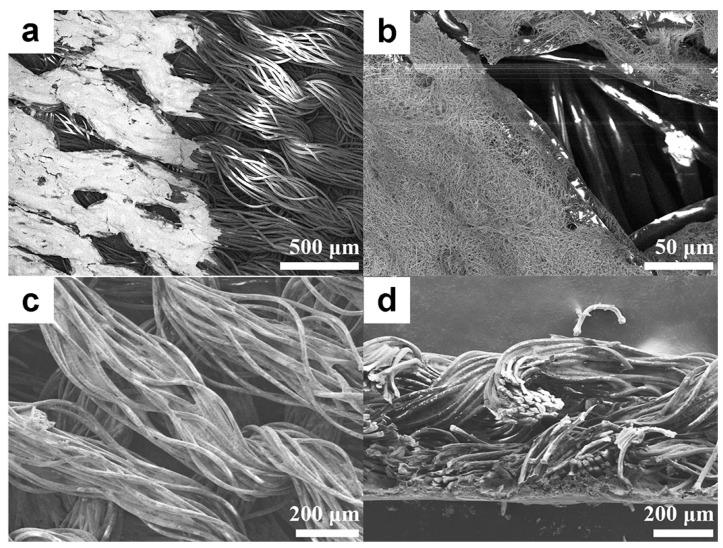
The SEM image of the interface between printed and un-printed areas (**a**), SEM image of the defect of printed area at the beginning area of printed tracks (**b**), top-view (**c**) and cross-view (**d**) SEM images of the textile electrode.

**Figure 5 nanomaterials-09-00686-f005:**
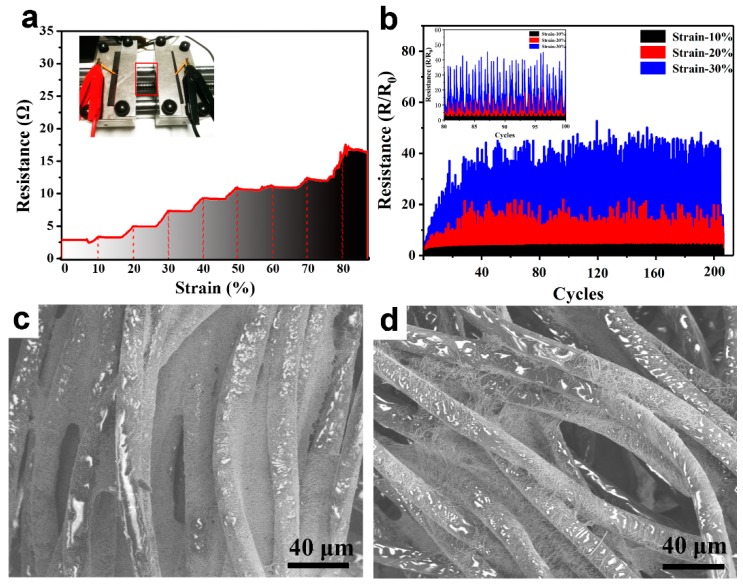
The resistance changes of the textile electrode with tensile strain (0–80%) and the insert image is the photograph of tensile test (**a**); the cyclic strain test if the textile electrode and the resistance change is cyclically stretched 200 times under the tensile strain of 10%, 20% and 30% (**b**, the insert is the magnification of the data); SEM images of the surface of the textile electrode before stretching (**c**) and after 80% tensile strain (**d**).

**Table 1 nanomaterials-09-00686-t001:** The conductive performance in different screen printing inks.

Conductive Contents (wt %)	Sintering Temperature	Printed Substrate	Sheet Resistance	Ref.
Ag flakes (70%)	850 °C	ferrite	0.022 Ω/sq	[26]
Ag NPs (60%)	200 °C	Glass	0.156 Ω/sq	[27]
Ag NPs (80%)	450 °C	alumina	0.006 Ω/sq	[28]
Ag flakes (70%)	120–180 °C	PET/Paper/PI	0.13 Ω/sq	[29]
Ag flakes (70%)	875 °C	LTCC	0.004 Ω/sq	[20]
Ag flakes (75%)	160 °C	textile	0.06 Ω/sq	[30]
Ag flakes (70%)	700 °C	alumina	0.009 Ω/sq	[31]
Ag NWs	160 °C	TPU	2.1 Ω/sq	[32]
Ag NWs	Room temperature	Glass	9.2 Ω/sq	[33]
Ag NWs (0.36%)	120 °C	PET	15.2 Ω/sq	[34]
Ag NWs (0.4%)	110 °C	PET	29.5 Ω/sq	[35]
Ag NWs (3.7%)	25 °C	Textile/paper	1.5/0.7 Ω/sq	Present work

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
