# Peer review of "Printing the Ultra-Long Ag Nanowires Inks onto the Flexible Textile Substrate for Stretchable Electronics"

_nanomaterials, 2019, doi:10.3390/nano9050686_

Reviewer 1 Report

The issues from the previous round of revision have been solved, and the work appears to be suitable for publication. 

Author Response

The issues from the previous round of revision have been solved, and the work appears to be suitable for publication. 

Responses: Thanks for reviewers’ support.

Reviewer 2 Report

The authors prepared silver nanowire inks using PVP, absolute ethanol and pure water and silver nanowire inks showed that low sheet resistance and high tensile resistance. Although the stability of the prepared ink is important in practical use, it is desirable to have a comment on that point. In addition, silver nanowires are easily corroded by hydrogen sulfide and sulfur compounds, but it is preferable that the description be given if this point is improved by PVP. Consideration for significant figures is desirable. The molecular weight of the polymer has no significant figure of six digits, and the electrical conductivity also has no significant figure of five digits.

Author Response

The authors prepared silver nanowire inks using PVP, absolute ethanol and pure water and silver nanowire inks showed that low sheet resistance and high tensile resistance. Although the stability of the prepared ink is important in practical use, it is desirable to have a comment on that point. In addition, silver nanowires are easily corroded by hydrogen sulfide and sulfur compounds, but it is preferable that the description be given if this point is improved by PVP. Consideration for significant figures is desirable. The molecular weight of the polymer has no significant figure of six digits, and the electrical conductivity also has no significant figure of five digits.

Responses: Thanks for reviewers’ comments. We agree with the stability of the prepared ink is important in practical use, we rechecked the as-prepared Ag NW ink before the submitted this manuscript, no any layered phenomenon can be observed because of the prepare ink is storage in the refrigerator. We have added the related description in our revised manuscript. For avoiding the Ag NWs are corroded by hydrogen sulfide and sulfur compounds, the insulate inks (e.g. green oil) are often used to protect the printed tracks in the practical application. In addition, the used PVP is purchase from the Sigma-Aldrich, and its CAS number is 9003-39-8. And its average mol. wt. is 360000, the detail information can be found in https://www.sigmaaldrich.com/catalog/product/vetec/v900010?lang=zh&region=CN. In our revised manuscript, we have changed the electrical conductivity description as “4.67×104 S/cm” for clarification.

This manuscript is a resubmission of an earlier submission. The following is a list of the peer review reports and author responses from that submission.

Round  1

Reviewer 1 Report

The work entitled, "Printing the Ultra-long Ag Nanowires Inks onto the Flexible Textile Substrate for Stretchable Electronics", talks about extra-long silver nanowire inks for screen printing and for coating on textile for stretchable and flexible electronics. 

Ag inks are already commercially available with water and solvents and various rheology modifies including but not limited to Polyurethanes and Polyacrylic acid etc. Commercially available Ag inks achieve sheet resistance as low as 0.1 ohms/sq for a thickness in excess of 10 microns. Therefore, thickness of the film is an important parameter when reporting the sheet resistance. In the current study a sheet resistance of 1.5 ohms/sq is reported, which is at least an order of magnitude lower than state of the art silver inks. Why is that so? what thickness is this for? 

The ink is also coated on textile and is used to measure change in resistance with strains (10-30%). There is a change of resistance as much as 400% relative to original resistance with a tensile strain of 30%. How do these results compare to some other works done on graphene based coating on textiles for strain sensing (Samad, Yarjan Abdul, et al. "From sewing thread to sensor: Nylon® fiber strain and pressure sensors." Sensors and Actuators B: Chemical 240 (2017): 1083-1090.).

Author Response

Responses to comments by Referee #1

1. Ag inks are already commercially available with water and solvents and various rheology modifies including but not limited to Polyurethanes and Polyacrylic acid etc. Commercially available Ag inks achieve sheet resistance as low as 0.1 ohms/sq for a thickness in excess of 10 microns. Therefore, thickness of the film is an important parameter when reporting the sheet resistance. In the current study a sheet resistance of 1.5 ohms/sq is reported, which is at least an order of magnitude lower than state of the art silver inks. Why is that so? what thickness is this for?

Responses: Thanks for reviewers’ comments. We agree that the conductivity of commercial Ag inks is higher than the current Ag NWs inks. However, all the commercial Ag inks require the sintering treatment after printing, and the sintering temperature is generally higher than 100 °C. Additionally, the commercial Ag inks typically consist of silver flake and resin, which is not suitable for printing on textile or other stretchable substrate materials. Herein, Ag NWs conductive ink is printed onto the surface of the stretchable textile and paper by facile screen printing technology, , the square resistance is 1.5 Ω/sq and 0.7 Ω/sq under natural drying conditions, respectively. Because of we use the screen printing method to fabricate the printed electrode, the thickness is about 6 microns.

2. The ink is also coated on textile and is used to measure change in resistance with strains (10-30%). There is a change of resistance as much as 400% relative to original resistance with a tensile strain of 30%. How do these results compare to some other works done on graphene based coating on textiles for strain sensing (Samad, Yarjan Abdul, et al. "From sewing thread to sensor: Nylon® fiber strain and pressure sensors." Sensors and Actuators B: Chemical 240 (2017): 1083-1090.).

Responses: Thanks for reviewers’ comments. We have read the mentioned article, the strain sensing performance is better than our works. Drop casting method is placed a droplet onto the substrate and wait for the solvent to evaporate, the obtained film thickness is depended on the concentration of solution. It is neither repeatable nor produces uniform films, it limits you to small samples and it takes a lot longer [1]. However, the purpose of our study is to prepare the Ag conductive ink rather than the strain sensor, and the textile conductive electrode is fabricated by printing our formulated inks. Printing method is a powerful technique to enable the production of large-scale, low-cost electronic devices and systems. This simple and cost-effective approach could enhance current methods of constructing a patterned surface for nanomaterials and offer opportunities for developing fully-printed functional devices [2].

Reference:

[1] Wei Wu, Stretchable electronics: functional materials, fabrication strategies and applications, Science and Technology of Advanced Materials, 2019, DOI: 10.1080/14686996.2018.1549460.

[2] Wei Wu, Inorganic Nanomaterials for Printed Electronics: A Review, Nanoscale, 2017, 9, 7342 – 7372.

Reviewer 2 Report

"Printing the Ultra-long Ag Nanowires Inks onto the Flexible Textile Substrate for Stretchable Electronics" submitted by Sheng-Hai Ke et al. reports the demonstration of Ag nanowire based conductive textile which is fabricated through screen printing, and the reviewer would like to ask for a revision for more complete manuscript. Some detailed comments are:

1. The shear rate has been changed from 0.1 s-1 to 200 s-1 and recovered to 0.1 s-1 for the measurement of the rheological properties of the ink, however, the background for using these specific values is not described in the manuscript. 

2. In Figure 4, the authors state that the ink penetrates into the gap between the fabric fibers, but the SEM image in Figure 4 does not show such phenomenon clearly. It might help if the authors can provide the SEM image of the textile electrode before the penetration of the Ag NW ink for comparison. SEM images at different magnification might help as well. 

3. The reviewer is not certain if the standard criteria for such textile electrode should be Ω/sq, since the configuration cannot be considered as a thin-film electrode. 

4. For the cyclic test in Figure 5b, higher magnification graph also should be included, at least for a few number of cycles.

5. The increase in resistance due to the strain seems to be fairly high (e.g. R/R0~40 for 30% strain after 200 cycles) compared to other studies on the stretchable electrodes. It would be helpful if the authors can kindly introduce some possible methods to prevent the resistance change. 

Author Response

Responses to comments by Referee #2

1. The  shear rate has been changed from 0.1 s-1 to 200 s-1 and recovered to  0.1 s-1 for the measurement of the rheological properties of the ink,  however, the background for using these specific values is not described  in the manuscript.

Responses: Thanks for reviewers’ comments. Different printing methods require different shear rate, because of the current printing method is screen printing, the shear rate range of 0.1 s-1 to 200 s-1 is simulated the ink transfer process during the screen printing. This value is reported in the previous literatures [1-3], we have added related description in our revised manuscript.

References:

[1] Tian B., Yao W., Zeng P., Li X., Wang H., Liu L., Feng Y., Luo C., Wu W., All-printed, low-cost, tunable sensing range strain sensors based on Ag nanodendrite conductive inks for wearable electronics. Journal of Materials Chemistry C, 2019, 7 (4), 809-818.

[2] Faddoul R., Reverdy-Bruas N., Blayo A., Formulation and screen printing of water based conductive flake silver pastes onto green ceramic tapes for electronic applications. Materials Science and Engineering: B, 2012, 177(13), 1053-1066.

[3] Liang J., Tong K., Pei Q., A water‐based silver‐nanowire screen‐print ink for the fabrication of stretchable conductors and wearable thin‐film transistors. Advanced Materials, 2016, 28(28), 5986-5996.

2. In  Figure 4, the authors state that the ink penetrates into the gap  between the fabric fibers, but the SEM image in Figure 4 does not show  such phenomenon clearly. It might help if the authors can provide the  SEM image of the textile electrode before the penetration of the Ag NW  ink for comparison. SEM images at different magnification might help as  well.

Responses: Thanks for reviewers’ comments. We have added the SEM image (the interface of printed and un-printed area and the defect printed area at the beginning area of printed tracks, as the Figure 4a and 4b) at higher magnification in our revised manuscript for clarification.

3. The  reviewer is not certain if the standard criteria for such textile  electrode should be Ω/sq, since the configuration cannot be considered  as a thin-film electrode.

Responses: Thanks  for reviewers’ comments. The conventional resistance of printed  electrode is related to the length of printing tracks, and the surface  of textile is very rough, resulting in the thickness and final  resistance of Ag NW printing layer cannot be obtained. The sheet  resistance is a very conventional parameter to characterize the  conductivity of electrode [1-3], and it can be measured by using the  four-point method.

References:

[1] Lv, J., Zhou, P., Zhang, L., Zhong, Y., Sui, X., Wang, B., Chen, Z., Xu, H., Mao, Z., High-performance textile electrodes for wearable electronics obtained by an improved in situ polymerization method. Chemical Engineering Journal, 2019, 361, 897-907.

[2] Liu, M., Pu, X., Cong, Z., Liu, Z., Liu, T., Chen, Y., Fu, J., Hu, W., Wang, Z. L., Resist-Dyed Textile Alkaline Zn Microbatteries with Significantly Suppressed Zn Dendrite Growth. ACS Applied Materials Interfaces, 2019, 11 (5), 5095-5106.

[3] Sun, P., Qiu, M., Li, M., Mai, W., Cui, G., Tong, Y., Stretchable Ni@NiCoP textile for wearable energy storage clothes. Nano Energy, 2019, 55, 506-515.

4. For the cyclic test in Figure 5b, higher magnification graph also should be included, at least for a few number of cycles.

Responses: Thanks for reviewers’ comments. Higher magnification graph of figure 5b have been added into the revised manuscript.

5.  The increase in resistance due to the strain seems to be fairly high (e.g. R/R0~40 for 30% strain after 200 cycles) compared to other studies on the stretchable electrodes. It would be helpful if the authors can kindly introduce some possible methods to prevent the resistance change.

Responses: Thanks for reviewers’ comments. It is very difficult to prevent resistance changing of printing electrode when textile is stretching, because of a part of the conductive networks is broken and the distance between the conductive fillers is increased [1-2]. The resistance cannot be fully recovered when released to 0%, which may be due to the reorientation and the increase in the initial distance of the conductive fillers [3-4]. Our previous work has reported a method that the conductive fillers are embedded into the flexible substrate, and it can prevent the resistance changing to some extent [5].

References:

[1] Liu, H., Li, Y., Dai, K., Zheng, G., Liu, C., Shen, C., Yan, X., Guo, J., Guo, Z., Electrically conductive thermoplastic elastomer nanocomposites at ultralow graphene loading levels for strain sensor applications. Journal of Materials Chemistry C, 2016, 4 (1), 157-166.

[2] Liu, X., Lu, C., Wu, X., Zhang, X., Self-healing strain sensors based on nanostructured supramolecular conductive elastomers. Journal of Materials Chemistry A, 2017, 5 (20), 9824-9832.

[3] Wang, S., Zhang, X., Wu, X., Lu, C., Tailoring percolating conductive networks of natural rubber composites for flexible strain sensors via a cellulose nanocrystal templated assembly. Soft Matter, 2016, 12 (3), 845-852.

[4] Hu, Y., Zhao, T., Zhu, P., Zhu, Y., Shuai, X., Liang, X., Sun, R., Lu, D. D., Wong, C.-P., Low cost and highly conductive elastic composites for flexible and printable electronics. Journal of Materials Chemistry C, 2016, 4 (24), 5839-5848.

[5] Zhang, S., Li, Y., Tian, Q., Liu, L., Yao, W., Chi, C., Zeng, P., Zhang, N, Wu, W., Highly conductive, flexible and stretchable conductors based on fractal silver nanostructures. Journal of Materials Chemistry C, 2018, 6 (15), 3999-4006.

Reviewer 3 Report

The paper 'Printing the Ultra-long Ag Nanowires inks onto the flexible textile substrate for stretchable electronics' by Ke S.H. et al. reports the preparation of high-aspect ratio Ag nanowires for application on textile electronics. Although the topic is really appealing for the scientific community, the paper in the present form cannot be published.

The manuscript lacks of novelty and the results presented are incomplete and not new.

Firstly, the authors did not describe adequately the state of the art about AgNW inks and their formulation is compared just with other conductive inks based on Ag-flakes or AgNPs which in my opinion is not appropriate. For example, in the paper 'Personal Thermal Management by Metallic Nanowire-Coated Textile' by Hsu P.C. at al. (NanoLetters, 2015) a similar concept is described but the authors did not mention this work.

Secondly, the authors claim the low temperature and the low AgNW content being the most notable parts of their results but, the paper 'A water-based silver nanowire ink for large-scale flexible transparent conductive films and touch screens' by Chen S. et al. reports better performance; I consider appropriate to add a reference even if no textile has been used as support.

Finally, the fabrication of the AgNW-based cloth which is the core of the work is not properly described and its characterization is not sufficient (the morphological characterization is poor and the sheat resistance does not even have an associated error).

The paper is similar to 'A Water-Based Silver-Nanowire Screen-Print Ink for the Fabrication of Stretchable Conductors and Wearable Thin-Film Transistors' (ref 16) with the only exception of the PVP additive which is replaced by a derivative of cellulose.

In my opinion the present work does not present a sufficient degree of novelty and cannot be consider for publication in Nanomaterials.

An extensive English editing is also required.

Author Response

Responses to comments by Referee #3

1. Firstly, the authors did not describe adequately the state of the art about Ag NW inks and their formulation is compared just with other conductive inks based on Ag-flakes or Ag NPs which in my opinion is not appropriate. For example, in the paper 'Personal Thermal Management by Metallic Nanowire-Coated Textile' by Hsu P.C. at al. (Nano Letters, 2015) a similar concept is described but the authors did not mention this work.

Responses: Thanks for reviewers’ comments. We have read this article and cited it in our revised manuscript, and added the comparison with other Ag NW ink for clarification. In the mentioned article, the CNT-cloth are fabricated by dip coating method, the textile is completely immerse in the Ag NWs dispersion and then vacuum-dried. In the dip coating, the substrate is dipped into the solution and then withdrawn at a controlled speed. Thickness determined by the balance of forces at the liquid-substrate interface. It is neither repeatable nor realize uniform films, it is often applied to fabricate the small samples and it takes a lot longer time to dry [1]. However, the purpose of our study is to prepare the Ag NW conductive ink, and the textile conductive electrode is fabricated by printing our determined inks. Screen printing method is a powerful technique to enable the production of large-scale, low-cost electronic devices and systems. This simple and cost-effective approach could enhance current methods of constructing a patterned surface for nanomaterials and offer opportunities for developing fully-printed functional devices [2].

Reference:

[1] Wei Wu, Stretchable electronics: functional materials, fabrication strategies and applications, Science and Technology of Advanced Materials, 2019, DOI: 10.1080/14686996.2018.1549460.

[2] Wei Wu, Inorganic Nanomaterials for Printed Electronics: A Review, Nanoscale, 2017, 9, 7342 – 7372.

2. Secondly, the authors claim the low temperature and the low Ag NW content being the most notable parts of their results but, the paper 'A water-based silver nanowire ink for large-scale flexible transparent conductive films and touch screens' by Chen S. et al. reports better performance; I consider appropriate to add a reference even if no textile has been used as support.

Responses: Thanks for reviewers’ comments. The reference have been cited in our revised manuscript.

3. Finally, the fabrication of the Ag NW-based cloth which is the core of the work is not properly described and its characterization is not sufficient (the morphological characterization is poor and the sheet resistance does not even have an associated error).

Responses: Thanks for reviewers’ comments. We agree this comment, and we have added the SEM image (the interface of printed and un-printed area and the defect printed area at the beginning area of printed tracks, as the Figure 4a and 4b) at higher magnification in our revised manuscript for clarification. In addition, the conventional resistance of printed electrode is related to the length of printing tracks, and the surface of textile is very rough, resulting in the thickness and final resistance of Ag NW printing layer cannot be obtained. We have added the magnification image of the change of resistance in Figure 5.

4. The paper is similar to 'A Water-Based Silver-Nanowire Screen-Print Ink for the Fabrication of Stretchable Conductors and Wearable Thin-Film Transistors' (ref 16) with the only exception of the PVP additive which is replaced by a derivative of cellulose.

Responses: Thanks for reviewers’ comments. The mentioned paper mainly describes a water-based silver-nanowire ink are printed onto PET/Glass/PUA for electronic applications. The sintering/curing temperature of Ag NW(6.6 wt%) ink prepared using a derivative of cellulose is 150°, but our Ag NW(3.7 wt%) ink prepared using PVP is room temperature, which is very suitable for the temperature-sensitive substrates of flexible electrodes. In addition, because of the surfaces of PET/Glass/PUA are very smooth, the printability of smooth substrate is generally well than the rough surface of substrates. In contrast, the surface of our used textile is very rough while its surface has a large amount of gaps between the fibers, which is a great challenge to prepare stretchable conductive electrode by screen printing directly. We envision our inks can be used for the temperature-sensitive substrates and fabricate the stretchable textile electrode by facile screen printing method.

5. In my opinion the present work does not present a sufficient degree of novelty and cannot be consider for publication in Nanomaterials. An extensive English editing is also required.

Responses: Thanks for reviewers’ comments. The novelty of our work is that we used the PVP as binder and water and absolute ethanol as solvent, a low silver content of Ag NWs conductive ink for screen painting was prepared, and successfully printed Ag NWs conductive ink onto the surface of stretchable fabric and paper. Indeed, the Ag NWs conductive ink has low conductive component content, good electrical conductivity after curing at room temperature and its printing electrode can maintain the conductivity when it is stretched. I have checked the English grammar in our revised manuscript again and the corrections are marked in red color

Round  2

Reviewer 3 Report

The main advantage of your work is the low sintering temperature of the ink which results in an impressive low resistance. In my understanding, your support/cloth has to be dried after electrodes patterning; I would like to know how long the drying process takes and how it can be advantageous from an industrial point of view.

Regarding the printing process, what is your electrode resolution on glass and on the cloth?

However I have to point out that the present work lacks of novelty and it is similar to other reports on AgNW/PVP conductive electrodes (exemple:'Very long Ag nanowire synthesis and its application in a highly transparent, conductive and flexible metal electrode touch panel' by Lee et al. 2012 Nanoscale).

Furthermore, an extensive english editing is still required.
